# Facilitating Factors of Professional Health Practice Regarding Female Genital Mutilation: A Qualitative Study

**DOI:** 10.3390/ijerph17218244

**Published:** 2020-11-08

**Authors:** M Idoia Ugarte-Gurrutxaga, Brígida Molina-Gallego, Laura Mordillo-Mateos, Sagrario Gómez-Cantarino, M. Carmen Solano-Ruiz, Gonzalo Melgar de Corral

**Affiliations:** 1Department of Nursing, Physical and Occupational Therapy University of Castilla-La Mancha, Campus Toledo, 13001 Ciudad Real, Spain; Maria.Ugarte@uclm.es (M.I.U.-G.); Brigida.Molina@uclm.es (B.M.-G.); Sagrario.Gomez@uclm.es (S.G.-C.); Gonzalo.Melgar@uclm.es (G.M.d.C.); 2Faculty of Health Sciences, Universidad de Castilla la Mancha, 45600 Talavera de la Reina (Toledo), Spain; 3Department of Nursing, University of Alicante, 03690 Alicante, Spain; carmen.solano@ua.es

**Keywords:** female genital mutilation, healthcare professionals, nursing, health

## Abstract

*Introduction:* According to figures released by UNICEF (United Nations Children’s Fund), more than 200 million girls and women have suffered female genital mutilation (FGM) in 30 African and Middle East countries. An increasing number of African women who come from ethnic groups where FGM is practised are arriving in Western countries. Healthcare professionals play a fundamental role in its prevention. *Goals:* To learn about the factors that healthcare professionals consider as facilitators for prevention and action when faced with female genital mutilation. *Methods:* A cross-sectional descriptive study developed on the basis of the qualitative methodological perspective, where 43 healthcare professionals participated. A series of analysis dimensions were established, based on which, the interview and discussion group scripts were designed. *Results:* Addressing FGM requires a series of structural adaptations of the healthcare system that facilitate the recording and monitoring of cases, both for treatment and for prevention. In addition, it is necessary to establish coordination between the healthcare, social services and education sectors. *Conclusions:* The existence of a protocol of action and training in its use is one of the key tools to take into account.

## 1. Introduction

More than 200 million girls and women alive today have been subjected to female genital mutilation (FGM) in the 30 countries in Africa, the Middle East and Asia where this practice is concentrated [1]. Close to 70,000 women from countries where FGM is practised are living in Spain, 18,000 of which are under the age of 14, i.e., those who run the risk of being mutilated when they travel on holidays with their families to their countries of origin.

FGM is defined by the World Health Organisation (WHO) [2] as “all procedures that involve partial or total removal of the external female genitalia, or other injury to the female genital organs for non-medical reasons”, and is recognised internationally as a violation of the human rights of girls and women.

The WHO [2] distinguishes four types of FGM: type 1—the partial or total removal of the clitoral glans and/or the prepuce; type 2—partial or total removal of the clitoral glans and the labia minora, with or without removal of the labia majora; type 3—the narrowing of the vaginal opening through the creation of a covering seal, where the seal is formed by cutting and repositioning the labia minora or labia majora, sometimes through stitching, with or without removal of the clitoral prepuce; type 4—all other harmful procedures to the female genitalia for non-medical purposes, e.g., pricking, piercing, incising, scraping and cauterizing the genital area. In the year 2016, several subcategories were added

The real situations of risk that we found were related to the stance of the parents and the family in the country of origin with regard to this practice and the moment when they travel to these countries during holiday periods.

FGM has serious consequences for physical, mental and social health [3,4,5].

The most immediate consequences are haemorrhages, gynaecological urinary infections, urinary retention and vaginal fistulas [6].

In the medium term, we found sexual problems, such as anorgasmia, pain during intercourse, alteration of sexual sensitivity, a reduction of sexual desire [7,8,9], chronic infections, infertility and childbirth complications [3].

Psychological consequences include post-traumatic stress disorder (PTSD), night terrors and psychosomatic illnesses, with symptoms such as tachycardia, pain or chest oppression, vomiting, muscular pain and diarrhoea [10].

The social consequences are related to the rejection the girl or woman may suffer. These consequences occur both in the host country (the female survivor of FGM is stigmatised) and the country of origin (women that have not gone through this procedure are discriminated against) [11].

During recent years, many destination countries have implemented laws, policies and programmes for both preventing and acting in the face of FGM [12,13].

In Spain, in the year 1993, the first cases began to be detected by healthcare professionals. Starting from that moment, both state and social actors have started to intervene in this sphere in order to facilitate the early detection of situations of risk and the prevention of FGM. At a legal level, FGM has been defined as a crime in the Criminal Code since 1995 (article 149.2) and is punished with prison sentences of 6–12 years and the revocation of legal custody in the case of minors. In parallel, a process to prepare protocols in autonomous regions was activated, seeking to comprehensively address this phenomenon in order to reinforce prevention from multiple perspectives, including legal, social, healthcare and educational.

Despite these initiatives, many survivors of FGM state that they have had negative experiences with the provision of healthcare [14,15,16], which was mainly associated with a lack of structural adaptation by the healthcare systems (translation, coordination, etc.) and a lack of cultural skills among healthcare professionals. Women felt discriminated against, stigmatised and in a situation of severe vulnerability [11].

In addressing these situations, healthcare professionals play a fundamental role, as they provide specific care for women and girls in both primary healthcare and specialised healthcare throughout their lives.

For this reason, the aim of our study was to learn about the factors that healthcare professionals consider as facilitators for prevention and action when faced with female genital mutilation.

## 2. Materials and Methods

### 2.1. Design of the Study

Twenty semi-structured in-depth interviews and 3 discussion groups were conducted, in which 43 health professionals with the following profiles participated: nurses, midwives, family medicine, pediatrics and obstetrics–gynecology.

All interviews were carried out by telephone after consulting the availability and receipt of the informed consent signed by the person to be interviewed by email. The discussion groups were held in person at the participants’ work centers. Table 1 shows an outline of the contents addressed in the interviews and in the discussion groups.

In order to carry out this study, we included a qualitative methodological perspective in order to learn about discourses, opinions and underlying ideas. It contained everything that was said and how it was said. The objectives of our study were focused on attitudes and beliefs, and therefore, this required research that was designed with a certain degree of openness and flexibility, enabling the emergence and recording of meanings and not statistical frequency.

This is why we chose a qualitative structural technique that identified realities with explanatory relevance. In short, we used a methodology that enabled us to collect discourses in all dimensions and to link them to the social conditions in which these discourses were produced, assigning meaning to them. The representativity of our sample did not lie in the “quantity” of the sample, but in guaranteeing the collection of the possible subjective configurations (values, beliefs, motivations, expectations and knowledge) of the community that was the subject of our study with respect to FGM.

We applied a qualitative content analysis approach, which as a method, is a systematic, objective and flexible resource for understanding a phenomenon that involves labeling and interpreting the data in their own context [17].

This study was part of a broader project funded by the Castilla-La Mancha Healthcare Service (SESCAM), Spain.

### 2.2. Sample and Configuration

Forty-three health professionals participated, both from primary care and hospital care, with the following profiles: nurses, midwives, family medicine, pediatrics and obstetrics–gynecology (Table 2).

### 2.3. Collection of Data

First of all, the interviewer introduced herself and provided information about the study and its goals. The average duration of the interviews was 60 min. In order to achieve a sampling with maximum variation, we selected key informers from both sexes and different profiles and fields of work (Table 2).

In the design of the discussion groups, we guaranteed the necessary heterogeneity within the homogeneity conferred by belonging to the same population group. Our aim was that the discourses of the groups confirmed that the information we collected was not determined by the characteristics of a specific group, but can be extrapolated to other groups with a similar make-up.

We subjected the number of discussion groups to the “data saturation criterion”. Theoretical saturation is reached when the information collected does not contribute anything new to the development of the properties and dimensions of the analysis categories. The criteria to determine saturation are as follows: (a) the integration and density of the theory (it is saturated when the highest number of variations within the theory have been analysed and explained, and when the relationship between the emerging categories obeys a logical-explanatory pattern of the problem researched), (b) the combination of the empirical limits of the data (saturation is reached when the researcher does not have access to other data that contributes towards the development of the research) and (c) the theoretical sensitivity of the person carrying out the analysis (the capacity of the researchers to address the data theoretically) [18].

An exhaustive recording of the qualitative data was carried out in terms of the transcription of the recordings during the discussion groups and interviews, as well as the recording of the non-verbal information collected by the observers.

### 2.4. Analysis of the Data

Later, by reading and encoding all the information (Atlas-Ti program, Scientific Software Development GmbH, Berlín, Alemania), the main dimensions around which the discourse is structured were identified, followed by a distinction between the most relevant aspects of each of the topics, by grouping the data collected around categories related to the specific objectives of the study.

The encoding and categorisation were verified by all members of the research team in order to reach a consensus. We also engaged in collaboration with an external reviewer with experience in the analysis of conventional content to verify the data encoding process, interpretation and categorisation. The collection of data was carried out from September to December 2019 in order to have sufficient commitment to the data [19].

### 2.5. Ethical Considerations

We received a favourable decision from the clinical research ethics committee of the Integrated Healthcare Department of Talavera de la Reina (Toledo, Spain) for this study (CElm Code: 37/2019, of 11 October 2019). The research did not involve any risk for the participants, as the techniques used and the contents of the study did not entail risks for their physical or psychological integrity.

Before starting the interviews and the discussion groups, there was an explanation of the study, its objectives, the possible applications, the importance of their (voluntary) participation and the confidentiality of the processing of the data collected, and they were informed that it was possible to withdraw from the study at any time. They were also asked if they were interested in receiving information about the results.

All the information collected was analysed in a confidential manner, guaranteeing the anonymity of the participants.

## 3. Results

The results are presented below and are structured according to the four topics and the emerging categories of the discourses of the participants in the study (Table 3).

### 3.1. Recording of Cases of FGM in the Clinical Record (CR)

#### 3.1.1. Benefits of Recording Cases of Female Genital Mutilation in the Clinical Record

The midwives who participated in the study commented that at the moment of childbirth, it may be adequate to record FGM in the woman’s CR due to the repercussion that it may have on her sexual and reproductive health, but they had doubts about where to locate the annotation, though they did not see any complications.

“*But there’s no specific place to… record it. At least that I know of. […] I don’t think it should be [...] complicated to record it correctly on an empty form*” (18:10 E6).

They commented that women who had suffered genital mutilation are more prone to tearing. In the CR, tears during childbirth were recorded, not their etiology, meaning that the FGM was not made visible.

“*Phew. My goodness! Well, they tear much more (emphasising). (...) Because, in the end, like everything… the whole perineum area is… is… is… [searching for the word] is scarred over. It’s scarred tissue. It doesn’t distend, and so, in the end, you have to perform wider episiotomies because otherwise… it breaks (stressing). All the tears are recorded in the Partogram*” (28:2 E15).

One of the primary healthcare paediatricians pointed out that it would be interesting to record the family’s country of origin in the girl’s CR during the development of the Child Health Programme. This would make it possible to address FGM preventively. Recording the nationality of the child’s family, although done voluntarily, could be done for other purposes such as guaranteeing continuity of the vaccination calendar.

“*It depends a bit on the… on the staff. But, for example, children… originally from Morocco (emphasises), for example, we warn them. When they are going to travel to Morocco, that they have to get the vaccine for hepatitis A, for example. It would be something similar. I mean, include in the record, in a section: Origin… And then, talk to the family. But of course, we need to know the country they’re from, don’t we? And then… ask the mother directly whether their population, whether their… community practices it or not*” (25:17 E10).

#### 3.1.2. Obstacles for Recording

##### Lack of communication between the management programs between the two levels of care 

There was a lack of communication between the management programs between the two healthcare levels.

The healthcare service of our field of research has a software program for managing the clinical records for each of the healthcare levels, namely, primary healthcare and specialised healthcare. These two programs were not connected; therefore, unless it was done intentionally by the healthcare professional, whatever was entered into the CR of the patient in a hospital was not automatically visible to primary healthcare.

There was no interaction between the entries in the clinical record done via the computer program used in specialised healthcare and that used in primary healthcare, and vice versa.

“*For example, a woman arrives who is 38 weeks pregnant, and automatically, when I open the record in the Mambrino software [SH], I cannot see the observations made at primary healthcare, and I have to go and ask*” (9:38 E2).

##### Omission due to prejudice and stereotypes: stigma

Despite the consensus on the appropriateness of recording FGM, there were those who were afraid to do so, possibly because it could stigmatise the woman who had been mutilated.

“*[m]aking highly visible the fact that a woman has suffered a sexual aggression, which is what a clitoral ablation is. Making it highly visible and clear in her record, I don’t know… I have my reservations regarding how much this could be harmful for her*” (26:15 E11).

### 3.2. Intersectoral Coordination

#### 3.2.1. Education Sphere

Although coordination between schools and healthcare services was non-existent, it was considered that the former can act both to educate families from countries at risk of FGM and to detect the risk of FGM upon learning about the family’s intention to travel, which is information they have access to and which should be communicated to the healthcare centre’s paediatric office.

“*I think that schools that have... pupils... who come from at-risk countries... well if they’re at-risk patients... I mean, if they’re children from at-risk countries, I think that, just like we inform the paediatricians, teachers could inform the… the families and see what happens. Even if they detect... the risk, also being able to be in touch with the paediatrician or the healthcare centre, but of course… These people sometimes don’t really go much to… to their healthcare centre*” (25:12 E10).

#### 3.2.2. Community Health Resources

In this section, we collect the discourses related to aspects that have to do with the existing/non-existent coordination between the different healthcare levels and between professionals with different healthcare profiles regarding how FGM was addressed.

There was a distinction made between abused women and women who had suffered genital mutilation. In the first case, a procedure had been established to notify the social worker; in the second, it had not.

“*Whenever a social intervention is necessary, we take care of cases at other levels; not mutilations, but abused women; we usually call the social worker and she takes care of... of the situation. I... have only seen one case of abuse where we realised during childbirth, because the woman had, well, she had bruising, she had... things, (lowers voice) and then we realised... during… during the childbirth, and the social worker was called. When it’s a social issue, what... what we do if we detect it is call the hospital’s social worker*” (27:11 E12).

It was considered that the coordination should be carried out between the professionals intervening in the healthcare process of the pregnancy, namely, the childbirth of women who were victims of GM (gynaecology and obstetrics, midwives, nurses and social workers) and those intervening in the healthcare for the girl (nursing and paediatrics).

“*Because, no matter how much awareness we have, paediatricians, the rest too because if a doctor for adults explores a woman and sees mutilation, he should know what to do then with respect to that, her sexual disorders or problems with childbirth, I mean, that’s because of something. But it’s fundamental that that information reaches the paediatrician, so that the paediatrician knows (emphasising) that the woman is mutilated, as it is a risk factor*” (17:12 E3).

This coordination requires a healthcare system structure that enables interaction between the two levels of healthcare provision and a protocol to systematise it. However, this coordination was only carried out informally, in what is called “corridor coordination”.

“*We do the coordination in the corridors. I’ve already established a mechanism. It’s this: every month I meet with the social worker to talk about these issues, but because I’ve established this with the social worker*” (12:61 GD2-GU).

In the discourses of the people interviewed, the mediator was identified as a key element to facilitate communication between women and healthcare professionals.

“*Very often they can’t understand you... I mean, we also need someone for support, a mediator*” (24:19 E14).

### 3.3. Training for Healthcare Professionals Regarding Female Genital Mutilation

#### 3.3.1. Deficiencies in Training

Most of the people who participated in this study stated their interest in receiving training to address FGM, though there were those who said that there may be a lack of knowledge about the existence of this practice in countries receiving immigration from countries at risk, and it was even said that this practice is associated with a cultural tradition and that addressing it is beyond the remit of healthcare professionals.

All of the people interviewed reported a lack of training in FGM, both in their academic education and after joining the workforce.

“*Rather I discovered it through... cases, the media even, or the training each one of us has, but... as such [...] training, at least when I did it (specialising in obstetric-gynaecological nursing, midwife), no… that topic wasn’t touched on*” (9:2 E2).

“*Personally, I think that we midwives are lacking training in this sense, in this specific issue... [...] (laughs gently) we have loads of training, but there’s a lack of knowledge about this… to the extent that we don’t know if it’s frequent or not, how frequent it is, whether there are… prevention programmes (emphasising), if there’s a moment when we can act, if… how we have to deal with it; I’ve already seen myself at the moment of childbirth when you discover there’s a mutilation… If there’s any way we can intervene… there’s a lack of knowledge, I think, where… we don’t know what to do*” (9:39 E2).

On the other hand, the consequences of not having received adequate training conditions them enormously in both the identification of cases of risk of FGM and in how to act when they discover that a woman has been mutilated. This created a lack of confidence and therefore they did not know whether to address it or not.

“*… We don’t know how we can prevent these things. Prevention is fundamental. And then the treatment... the treatment [...] once you...face a case like this...you don’t know how to react… whether you can do something, whether you can’t...If...for example, a pregnant woman, you don’t know whether she needs some kind of special care*” (9:40 E2).

It was also pointed out in the interviews that the lack of training may mean that cases of FGM go unnoticed.

“*No. I haven’t seen any. I don’t know whether...if I found a case whether I’d be capable of detecting it. Unless I was explicitly looking for it, of course. If you look for it explicitly, I imagine you would. But in our job, where...well, you go to fit a catheter, or you go to do…something. I don’t know if you’d be able to, without training*” (10:7 E4).

#### 3.3.2. The Contents of the Training

The aspects for which they considered they should receive training were as follows: countries where FGM is practised, identification of FGM risk factors for its prevention, communication skills and the need for training to provide culturally adequate care.

“*Not the fact of the mutilation itself, but what it entails at a cultural level, for example, that it’s... that... it’s.... We don’t know, there’s no knowledge. I mean, there’s talk sometimes about, well, the usual, that if they don’t mutilate the girl then their people reject them, by their tribes and their... villages. Well, those kinds of things... What... Everything it entails, not just the act itself, maybe yes, it’d be good to know more about it. And of course how to act. (Sound of a computer keyboard) Yes, yes, well... It’d be good*” (27:9 E12).

“*It might feel violent to them if I start broaching a subject that they haven’t talked to me about first. They’d have to come and tell me what’s happening, wouldn’t they? A direct, imposing approach, with another culture, I think they might even, with the conditioning factors involved, they can even back off and you lose the patient*” (11:16 GD1-GU).

### 3.4. Need for a Protocol to Address Female Genital Mutilation

#### 3.4.1. Benefits of a Protocol to Provide Care for Female Genital Mutilation

On the one hand, the professional advantages were mentioned, which helped or would help the professional in their healthcare work, and on the other hand, advantages were seen that would lead to mitigating the problem of FGM.

Protocols are useful as guidelines for action and orientation, without which, the healthcare workers are exposed to having their professional credentials questioned. This becomes much more important when dealing with complex matters that involve components that are traditionally removed from more classic competencies (biomedical).

Along these lines, for the participants, these types of cases appeared as “complicated” and “special”, particularly due to the psychological and social aspects. The problem itself even appeared as unknown in its basic epidemiological aspects.

“*We don’t know how to prevent it, nor do we know what to do. Whether they need… special attention, special care, or even a more psychological type of care, […] a person that has had this… this problem or this type of… of problem*” (9:13 E2).

Thus, a protocol emerged as something essential in the discourses because it helps (or would help those unaware of its existence) to learn about the problem and to organise the provision of care, even forcing the people responsible for the institution to take measures and act when they are required to by professionals.

“*I understand that, if they don’t have a protocol, well the nursing management won’t act, nor will they get involved. If we don’t know what to do. End of. Or: “It’s already done… so, if it’s already done, then what can we do, if it’s already… it’s been done… There’s no longer… There’s no going back. I don’t know. So, a protocol is absolutely necessary, of course*” (10:12 E4).

There was also a unanimous agreement that a protocol has (or would have) positive effects on the woman’s health.

“*I think it should be included because it’s something that must… have an impact on her sexual and reproductive health, of course, but there’s not […] very often there isn’t even an item to… to record it*” (9:29 E2).

Of course, it was considered that a protocol, including a systematic and appropriately integrated recording system between primary healthcare and specialised healthcare, would help a lot to organise healthcare provisions for FGM and its prevention. Unfortunately, integration between the computer records in primary healthcare and specialised healthcare also seemed difficult in this matter.

“*For example, a woman arrives who is 38 weeks pregnant, and automatically, when I open the record in the Mambrino software, I cannot see the observations made at primary healthcare… We often have to duplicate, or even triplicate information*” (9:38 E2).

In general, it was understood that a protocol unified at the SESCAM level that was obligatory (as it is for other problems with a strong legal component) would help to provide better healthcare and prevention, and help to raise visibility and awareness about the problem.

“*But anyway, they should have their protocol, learn about it, and make it compulsory. I mean, just like there’s a gender violence protocol, and when someone arrives at A&E or the surgery, there… there’s a protocol that’s followed, which they have and have to follow (emphasises), well they should also follow this protocol*” (10:14 E4).

## 4. Discussion

Our study highlighted the difficulty in recording FGM in the mother’s CR at the moment of childbirth due to a lack of adaptation of the system. This finding agrees with what we have found in other studies, where the systematic recording of FGM in the CR is very deficient in countries receiving immigration [13]. Several studies expose missed opportunities to diagnose FGM [20]. Taking into account that in 2016, a new code was added to identify FGM (code Z91.7) to the CIE-10 [21], it is surprising that it is not used to record it in the CR.

Another aspect that emerged in the discourse of the people who have participated in our study is concern about the possible stigmatisation of women in the event that their situation of having been mutilated is reflected in their CR. It could be that the lack of knowledge [22] and the emotional component present in professional interventions related to this practice can even lead to a paralysis, despite the enormous abuse of human rights, due to a desire to be sensitive to these women’s culture and to not impose “Western” culture. In addition, the lack of training in cultural skills could be behind this idea of stigmatisation that professionals hold, and which leads, as a result, to an infra-diagnosis of FGM. This concern that emerged in our study is shared by other research [23,24].

Training in FGM is a topic that concerns us, as in the community where we have carried out the research, there is a large number of people from at-risk countries. We know that women affected by FGM come into contact with different profiles of healthcare professionals, either during routine visits during the perinatal period or when monitoring health problems linked to female genital mutilation. On the other hand, girls who are the children of women who have been mutilated are guaranteed monitoring at the regular checkups that are scheduled within the Child Health Programme.

All of the people interviewed reported a lack of training in FGM, both in their academic education and after joining the workforce. This reality is shared with other countries where there is a significant number of people from at-risk countries [20,25,26].

In Spain, specifically in the region where our research was carried out, the presence of immigrants from countries where FGM is practised is becoming increasingly frequent. However, healthcare professionals do not receive training in this issue as part of their graduate and postgraduate education, or just receive scarce, voluntary training provided by Médecins du Monde. Nor do they seem to be aware that it is a public health problem and that, as such, it falls within their duty of care or the legislation in force in Spain or in the autonomous communities (Law for a Society Free from Gender Violence), where FGM is recognised as a form of gender violence and aims to provide training for healthcare professionals.

There is evidence that, despite the government’s efforts to eradicate the practice of FGM, there is a lack of training among healthcare professionals to address its treatment. We found this lack of training in the scientific literature to which we have access [27], which shows that the training strategies implemented in many of the countries that have declared their commitment to the eradication of FGM are not adequate.

The consequences of not having received adequate professional training very significantly limits the identification of cases at risk of FGM, the identification of women who have already been mutilated and the prevention of FGM among the daughters of these women.

In our study we found that when faced with a case of FGM, the woman is not asked due to a desire to not embarrass her, they find the situation uncomfortable. This passive attitude is largely conditioned by a lack of skills in solving situations where there is a huge clash of cultures and leads to missed opportunities for treatment, which has a negative impact on these women’s health [28].

Acquiring knowledge and an understanding of the culture of the women from these countries at risk of FGM is seen as one of the factors that would facilitate treatment the most. This aspect is also made evident in other research similar to ours [29,30]. Sensitivity, empathy and cultural humility improve interactions during the provision of medical care [31].

In our results, two sectors emerge, in addition to the healthcare sector, that are considered key for addressing FGM: education and social services. The importance of community health coordination when handling gender violence has been extensively documented, and the same occurs when the person who requires healthcare is in a situation of social vulnerability. In a systematic revision by Robertshaw, which aimed to learn about the challenges and facilitating factors for healthcare professionals who provide healthcare services for refugees and asylum seekers in their host countries [32], several studies were found showing that coordinating healthcare and social services could support the work of healthcare professionals and facilitate comprehensive care. In the study by Costello [33], particular mention was made of the social worker’s role as a key agent in this type of care. In addition, for the participants in our study, schools emerged as key agents for both the prevention and the monitoring of FGM in girls at risk. The obligatory nature of schooling means that the school has records of families’ journeys to their countries of origin (as a risk factor). This idea is shared by other studies [34]. Although not acknowledged in our study, the figure of the school nurse could be highly useful regarding prevention through education in sexual and reproductive health [35,36].

The demand for a protocol of action to address FGM emerged clearly from our research. This result is aligned with the current operation of the healthcare model where, for the sake of efficiency and effectiveness, a protocol is promoted for actions to be contextualised within the different healthcare levels and in each one of the issues addressed. In relation to FGM, several studies have defended the appropriateness of a protocol of action to address FGM [29].

It is paradoxical that, of the 43 people interviewed, none were aware of the existence of the protocol for the prevention of FGM that has existed in the region of the study since the year 2017, although it does not develop the guidelines to address the intercultural communication required when carrying out an interview with a woman that is at risk of or is susceptible to FGM.

## 5. Conclusions

There is evidence of great difficulty in recording FGM cases due to a number of reasons: it does not seem appropriate to record it at certain moments of healthcare provision (for example, during childbirth); there is no specific space where it can be recorded or it goes unnoticed (“it isn’t flagged”); only pre-established quantitative data are recorded, depending on the moment the healthcare is provided; even though it is recorded, the scarce interaction between the computer programs of specialised healthcare and primary healthcare prevents the monitoring of cases, meaning that they go unnoticed and are not recorded again; the etiology of the tear is not recorded during childbirth. Although it seems possible that a case of FGM can be included in the CR in such a way that, when opening it, it is “flagged” and the fear is that this measure might stigmatise the woman who is a victim of FGM.

The need to improve coordination between the different professional profiles that are most frequently linked to treatment for FGM, namely, midwives, paediatricians and gynaecologists, was recognized. Currently, an informal type of coordination is being carried out (“corridor coordination”).

Coordination with social and education services would facilitate the prevention of FGM among girls in a situation of risk. The role of teachers is key in the identification of certain risk factors (travel to the family’s country of origin). A possible relationship was seen between healthcare professionals and social services, similar to that which exists to address gender violence.

The figure of the intercultural mediator emerged as a key element in the process to prevent FGM.

The lack of training in FGM among the healthcare professionals in both their academic education and once they had joined the workforce conditioned them greatly, both in terms of the identification of cases at risk of FGM and how to act when they discover a woman has been mutilated. This creates a lack of confidence, and therefore, they do not know whether to address it or not.

Related training should provide skills regarding how to control the twofold discomfort produced when providing care: that of the women (due to the “stigma”, matters related to sexism, cultural differences and customs) and that experienced by the person treating them (such as the fact that it is “a delicate topic”, which is uncomfortable and difficult to deal with, with psychological repercussions for the professional, among other aspects).

The need for there to exist or to apply a standardised protocol was seen as essential, both by those who are familiar with the protocol and by those who are unaware of its existence.

Indeed, we can consider that establishing a protocol for many health problems not only contributes towards providing quality standards for healthcare provision but also towards safeguarding the professional who, by being able to adhere to what is established in specific protocols, would for all purposes be carrying out a task considered to be correct. Thus, protocols can be used as guidelines for action and orientation, without which, professionals are exposed to seeing their professional credentials questioned.

## Figures and Tables

**Table 1 ijerph-17-08244-t001:** Outline of the contents and the script of the interviews and the discussion groups.

Themes	Questions
Personal information	Sex Training Scope of work: AP/AE Service/work unit
Female genital mutilation (FGM) in the clinical record (CR)	Is FGM registered in the CR? Is there a space where I can make a record? Do you think it should be reflected?
Training on the approach to FGM	Do you know what FGM is? Do you know the consequences of the health of women who survive this cultural practise? Have you received training (undergraduate/postgraduate/work environment) on how to deal with a case of FGM? Do you think specific training is necessary?
Intersectorality in the approach to FGM	Do you think that the approach to FGM is solely the responsibility of the health system? What role do you think the educational field plays? What role do you think social resources play? Is there intersectoral coordination (education, social services and health)?
Usefulness of an action protocol before FGM	Do you think that the existence of an action protocol would facilitate healthcare for women who have been mutilated? Would it help to prevent FGM in girls?

Source: Own elaboration by the authors. Note: AP- primary healthcare, AE- specialised healthcare

**Table 2 ijerph-17-08244-t002:** Health professionals who participated in the study.

Professional Profile	Primary Care		Hospital Care		Total
	W	M	W	M	
Nurses	10		1		11
Family medicine	9	2			11
Pediatrics	5	2	3		10
Midwives	3		2	3	8
Gynecology–Obstetric			2	1	3
Total	27	4	8	4	43

Source: Own elaboration by the authors. Note: W—women, M—men.

**Table 3 ijerph-17-08244-t003:** Themes and categories identified after thematic analysis.

Themes	1	2	3	4
Facilitating factors for prevention and action against FGM	Record of FGM in the clinical record	Intersectoral coordination	Training for health professionals in FGM	Need for a protocol to address female genital mutilation
Categories emerging	Registry benefits	Educational field	Training gaps	Benefits of having a protocol
	Barriers to registration; miscommunication between management programs of the two healthcare levels; omission due to prejudices and stereotypes: stigma	Socio-health resources	Training contents	

Source: Own elaboration by the authors.

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
