# Peer review of "Facilitating Factors of Professional Health Practice Regarding Female Genital Mutilation: A Qualitative Study"

_ijerph, 2020, doi:10.3390/ijerph17218244_

Round 1
Reviewer 1 Report
This study reported the facilitating factors on professional health practice in female genital mutilation. Authors chose a qualitative-structural technique, including 43 healthcare professionals from the fields of primary healthcare and specialized healthcare (nursing, midwives, general practice, pediatrics, and gynecology-obstetrics. Findings of the study highlight that female genital mutilation requires different structural adaptation of healthcare system that facilitate the recording of the case as well as the monitoring of the case.
The aim of the article is interesting, but Authors should explain the method use to choose the 43 healthcare professionals. Are they representative of the healthcare in Spain?
Table 1: explain the meaning of W and M
Author Response
REVIEWER 1
Comments and Suggestions for Authors
This study reported the facilitating factors on professional health practice in female genital mutilation. Authors chose a qualitative-structural technique, including 43 healthcare professionals from the fields of primary healthcare and specialized healthcare (nursing, midwives, general practice, pediatrics, and gynecology-obstetrics. Findings of the study highlight that female genital mutilation requires different structural adaptation of healthcare system that facilitate the recording of the case as well as the monitoring of the case.
- The aim of the article is interesting, but Authors should explain the method use to choose the 43 healthcare professionals. Are they representative of the healthcare in Spain?
We have taken into account when selecting the professional profile of participants, those who at some point could get in contact with women who have undergone Female Genital Mutilation and / or girls at risk. Their scope of work was also taken into account: Primary Health Care (health centers) or Specialized Care (hospitals, specialty centers, outpatient clinics).
- Table 1 (New Table 2): explain the meaning of W and M
A note was added under Table 2 (line 117)

Reviewer 2 Report
This paper is a manuscript of a qualitative study about the factors that healthcare professionals consider as facilitators in women with female genital mutilation. I want to congratulate the authors for their effort and dedication in this study that addresses an interesting topic such as female genital mutilation. It is well written and understandable paper. However, I have some query that I would suggest clarification:
Materials and Methods
- (Line 107). It must be justificated why these healthcare professionals where choose, and why were not included others like specialising physiotherapies in women (obstetric and gynaecological). It is advisable to avoid starting a sentence with a number.
- (line 106). Which were the selection / inclusion criteria for healthcare professionals?
- Table 1 shows the results of the sample of healthcare professionals, it should be included in the Results section, not in Materials and Methods section.
- Table 2. How and who drew up the questions of the interviews?
- In all tables must be included the abbreviations used in a footnote to the table.
- (Line 137). How was recording the non-verbal information, clarify.
- (Line 145). Who made up the research team?
- Was a literal transcription of the recordings made?
Results
- How many discussion groups were formed? And how many semi-structured interviews were conducted?
- How many dimensions were identified?
- Who reviewed the results of the interviews?
- Who analyzed and coded the transcripts?
- Was any triangulation process carried out to ensure the validity of the results? In fact, triangulation is one of the tools qualitative research to add validity and quality to qualitative studies.
- Were the results analyzed based on the sex of the healthcare professionals? Could this data be relevant?
Discussion
- How difficult can it be for the interviews to carry out via telephone?, could it influence in the results?
Conclusion
- It would be recommend to abbreviate the conclusion.
Author Response
Materials and Methods
- (Line 107). It must be justificated why these healthcare professionals where choose, and why were not included others like specialising physiotherapies in women (obstetric and gynaecological). It is advisable to avoid starting a sentence with a number.
We have taken into account when selecting the professional profile of participants, those who at some point could get in contact with women who have undergone Female Genital Mutilation and / or girls at risk. Their scope of work was also taken into account: Primary Health Care (health centers) or Specialized Care (hospitals, specialty centers, outpatient clinics). In Castilla La-Mancha Health Service, there is no profile of a physiotherapist specialized in obstetric and gynecological health.
We change the beginning of the sentence to avoid doing it with a number(line 109). We thank the reviewer for the comment.
- (line 106). Which were the selection / inclusion criteria for healthcare professionals?
We have taken into account when selecting the professional profile of participants, those who at some point could get in contact with women who have undergone Female Genital Mutilation and / or girls at risk. Their scope of work was also taken into account: Primary Health Care (health centers) or Specialized Care (hospitals, specialty centers, outpatient clinics).
- Table 1 shows the results of the sample of healthcare professionals, it should be included in the Results section, not in Materials and Methods section.
The sampling strategy in qualitative research is the a priori determination, hence it should be incorporated in the Materials and Methods section in the study design.
- Table 2. How and who drew up the questions of the interviews?
The questions were written by the researchers. The formulation was subject to the relationship with the objectives of the study.
- In all tables must be included the abbreviations used in a footnote to the table.
Abbreviations were included in a footnote under Table 2 (line 117).
- (Line 137). How was recording the non-verbal information, clarify.
Non-verbal information was recorded by the observers (members of the research team) during the development of the discussion groups (face-to-face).
- (Line 145). Who made up the research team?
Conceptualisation, MI U G and L M M; methodology, MI U G, L M M and G M C; validation, MC S and B M G; formal analysis, MI U G and L M M; investigation, MI U G and G M C; writing—original draft preparation, , MI U G, L M M and G M C and B M G; writing—review and editing, L M M; supervision, G M C and S G-C. All authors have read the published version of the manuscript.
- Was a literal transcription of the recordings made? Yes, a literal transcription was made.
Results
- How many discussion groups were formed? And how many semi-structured interviews were conducted?
This information was included in the manuscript (line 83).
- How many dimensions were identified?
Four dimensions were identified and listed in Table 3: Record of FGM in the Clinical Record; Inter-sectoral coordination; Training for health professionals in FGM; Need for a Protocol to address Female Genital Mutilation (line 165)
- Who reviewed the results of the interviews?
The analysis of the content of the interviews (and of the Discussion Groups) was carried out by two researchers of the team (MI U G and L M M) and there was also an external reviewer participating.
- Who analyzed and coded the transcripts?
The entire analysis process (selection of verbatins, coding, establishment of categories, etc.) was carried out by two research the team (MI U G and L M M). The ATLAS.ti 8 Windows computer program (Scientific Software Development GmbH, Berlin, Germany) was used for coding.
- Was any triangulation process carried out to ensure the validity of the results? In fact, triangulation is one of the tools qualitative research to add validity and quality to qualitative studies.
To ensure the quality and validity of the results, we collected information through an iterative process of data collection and analysis through interviews and discussion groups until data saturation is reached under the criteria of three researchers.
- Were the results analyzed based on the sex of the healthcare professionals? Could this data be relevant?
In preparing the sample, it was guaranteed that there was a presence of men and women in all the profiles in which it was possible. The differentiated qualitative analysis between different variables (such as degree, scope, sex, etc.) was not carried out since it is not possible to obtain a representativeness of the speeches and it is not pursued.
Discussion
- How difficult can it be for the interviews to carry out via telephone?, could it influence in the results?
The telephone interview is a technique that allows information to be obtained in contexts where the geographic dispersion is great and there are difficulties to move. The research team has extensive experience in conducting interviews, both face-to-face and by telephone, and that was an important advantage. However, we consider that face-to-face interviews have a greater potential to create an environment of trust during their development.
Conclusion
- It would be recommend to abbreviate the conclusion.
We thank the reviewer for the observation. We reduced the length of this section while preserving the essentials.

Reviewer 3 Report
Facilitating factors on professional health practice in Female Genital Mutilation: a qualitative study.
Reviewer statement:
Female genital mutilation still affects million girls and women all over the world. Prevention of genital mutilation is very important. Although it is thought that genital mutilation is an African topic, but due to migration it also affects women in the Western countries. The authors conducted a study aiming to detect factors affecting Health care professionals in prevention and handling in women with genital mutilation, which is important and clinical relevant.
Title:
The title chosen reflect the study being reported and is adequate.
Overall:
It was difficult as a reader to read and understand the article. Although, the English grammar was good.
Abstract
Should be adjusted based on the provided comments.
Introduction
The introduction section is well written. The background of performing the current study is explained. There are some point needing clarification.
- The introduction section is far too long. The length can be reduced without losing the essentials. Please do so.
- The references are not in the good order, it start with reference 2 in the first sentence.
- In line 34 the word between should be removed.
- I would suggest to start the introduction section with the sentences on line 42 to 46.
- In line 64-64, the reference is not a number, but the authors. This should be changed.
Materials and methods
This section is well written, but it was a struggle to go through this section as a reader.
- The section study design does not provide sufficient information for the reader to understand the design and execution of the study. This should be changed. Only when reading collection of data, I realized that interviews were performed. The essentials of the methods are that the authors performed interviews with health care professionals, and in order to learn about attitudes and beliefs, performed a qualitative-structural methodology.
- In line 97 to 100 the authors report: “The representativity of our sample does not lie in the "quantity" of the same, but in guaranteeing the collection of the possible subjective configurations (values, beliefs, motivations, expectations and knowledge) of the community that is the subject of our study with respect to FGM.” The question remains how did the authors succeed to guarantee the collection of the possible subjective configurations in their approach? This is crucial and need explanation to the reader.
- In line 107 the authors report the inclusion of 43 health care professionals. This comes out of the blue, no explanation is given for the number of professionals and subsequently for the distribution of health care professionals. This is in line with the previous question, please elucidate extensively on this topic, as this is crucial for the interpretation of the results.
- Line 113-117 should be moved to the design of study section.
- In line 118-120 the authors report: “In order to achieve a sampling with maximum variation, we selected key informers from both sexes and different profiles and fields of work (Table 2). Could the authors inform the reader on the term key informers, what was meant and who were considered key informers?
- In line 123 the authors report the design of discussion groups. This was not reported in the study design section and as a reader it is unclear who, when and how this was done? As a reader I was lost in the design of this study, It still remains unclear how this study was executed. This is crucial and essential to be able to understand what how and was done and for the interpretation of the results. This essential point needs clarification.
- Furthermore, what kind of and how were the non-verbal information collected?
Results
This section is well written and good to understand. Nevertheless, some important points are missing or need clarification.
- The result are presented according to the four topics and emerging categories of the discourses of the participants. This is not clear to the reader, what was done with the other results, and were the participants the interviewed professionals?
As a reviewer it is not possible to review the results, discussion and conclusion section as the material and methods used to conduct this study are not clear, making it impossible to review these sections.
Author Response
- Should be adjusted based on the provided comments.
The modifications incorporated in the manuscript do not entail changes in the abstract.
Introduction
The introduction section is well written. The background of performing the current study is explained. There are some point needing clarification.
- The introduction section is far too long. The length can be reduced without losing the essentials. Please do so.
We made an effort to reduce the content of the Introduction section while preserving the essential aspects
- The references are not in the good order, it start with reference 2 in the first sentence.
We thank the reviewer for the comment. We already corrected it.
- In line 34 the word between should be removed.
The word “between” was removed.
- I would suggest to start the introduction section with the sentences on line 42 to 46.
Sentences were changed as suggested. Now in lines 30 to 34.
- In line 64-64, the reference is not a number, but the authors. This should be changed.
Thanks for the observation, it was a mistake as this reference shouldn’t be there.
The mistake was corrected.
Materials and methods
This section is well written, but it was a struggle to go through this section as a reader.
- The section study design does not provide sufficient information for the reader to understand the design and execution of the study. This should be changed. Only when reading collection of data, I realized that interviews were performed. The essentials of the methods are that the authors performed interviews with health care professionals, and in order to learn about attitudes and beliefs, performed a qualitative-structural methodology.
The information requested was added in the article in lines 83 to 85.
- In line 97 to 100 the authors report: “The representativity of our sample does not lie in the "quantity" of the same, but in guaranteeing the collection of the possible subjective configurations (values, beliefs, motivations, expectations and knowledge) of the community that is the subject of our study with respect to FGM.” The question remains how did the authors succeed to guarantee the collection of the possible subjective configurations in their approach? This is crucial and need explanation to the reader.
The professional profiles of the participants were determined by numerous variables (field, degree, sex, etc.) and were elaborated to guarantee a structural representativeness that reflected all the possible discursive positions in the field.
- In line 107 the authors report the inclusion of 43 health care professionals. This comes out of the blue, no explanation is given for the number of professionals and subsequently for the distribution of health care professionals. This is in line with the previous question, please elucidate extensively on this topic, as this is crucial for the interpretation of the results.
Table 2 (line 115) collects the profile of each participant and was prepared looking for a structural, not statistical, representativeness that would allow to collect all the possible existing discourses in the field. For this reason, all the profiles that a priori were estimated to have different discursive positions are included. To achieve saturation of the discourse in each of the topics, new participants (from previously established profiles) were included until the information was redundant and no new information was added.
- Line 113-117 should be moved to the design of study section.
Lines 83 to 90 were modified as suggested.
- In line 118-120 the authors report: “In order to achieve a sampling with maximum variation, we selected key informers from both sexes and different profiles and fields of work (Table 2). Could the authors inform the reader on the term key informers, what was meant and who were considered key informers?
We use the term "key informers" in the proper sense of qualitative anthropological research as those people who, due to their position in "the field", concentrate key information about the phenomenon to be studied.
- In line 123 the authors report the design of discussion groups. This was not reported in the study design section and as a reader it is unclear who, when and how this was done? As a reader I was lost in the design of this study, It still remains unclear how this study was executed. This is crucial and essential to be able to understand what how and was done and for the interpretation of the results. This essential point needs clarification.
This information was included in lines 83 a 85.
- Furthermore, what kind of and how were the non-verbal information collected?
In the development of the Discussion Groups carried out in person, the observers (members of the research team) took note of the gestures, tone of voice, interest of the participants ... revealing of the emotions that the topic discussed at each moment aroused. These notes were collected and taken into account in the analysis. Therefore, this allows incorporating the emotional dimension of the discourse and the factual dimension (what the participant intends to produce in the interlocutor) and incorporating it into the analysis.
Results
This section is well written and good to understand. Nevertheless, some important points are missing or need clarification.
- The result are presented according to the four topics and emerging categories of the discourses of the participants. This is not clear to the reader, what was done with the other results, and were the participants the interviewed professionals?
All results relevant to the study objectives were included in the analysis (selected and coded in each transcript). Those rejected correspond to issues not relevant to the investigation, circumstantial or accessory (issues under pressure, current healthcare issues, etc.).

Round 2
Reviewer 3 Report
The current manuscript is suitable for publication.